# Cytoplasmic Colocalization of RXRα and PPARγ as an Independent Negative Prognosticator for Breast Cancer Patients

**DOI:** 10.3390/cells11071244

**Published:** 2022-04-06

**Authors:** Wanting Shao, Melitta B. Köpke, Theresa Vilsmaier, Alaleh Zati Zehni, Mirjana Kessler, Sophie Sixou, Mariella Schneider, Nina Ditsch, Vincent Cavaillès, Udo Jeschke

**Affiliations:** 1Department of Obstetrics and Gynecology, University Hospital, LMU Munich, Marchioninistr. 15, 81377 Munich, Germany; schaowt111@126.com (W.S.); theresa.vilsmaier@med.uni-muenchen.de (T.V.); alaleh.zati@med.uni-muenchen.de (A.Z.Z.); mirjana.kessler@med.uni-muenchen.de (M.K.); 2Department of Breast Center, School of Medicine, The First Affiliated Hospital, Zhejiang University, Hangzhou 310003, China; 3Department of Gynecology and Obstetrics, University Hospital Augsburg, Stenglinstr. 2, 86156 Augsburg, Germany; melitta.koepke@uk-augsburg.de (M.B.K.); mariella.schneider@uk-augsburg.de (M.S.); nina.ditsch@uk-augsburg.de (N.D.); 4Faculté des Sciences Pharmaceutiques, Université Toulouse III—Paul Sabatier, F-31062 Toulouse, France; sophie.sixou@inserm.fr; 5IRCM-Institut de Recherche en Cancérologie de Montpellier, INSERM U1194, Université Montpellier, Parc Euromédecine, 208 rue des Apothicaires, F-34298 Montpellier, France

**Keywords:** breast cancer, nuclear receptor, RXRα, PPARγ, cytoplasmic expression, survival analyses

## Abstract

Retinoid X receptor α (RXRα) is a nuclear receptor (NR) which functions as the primary heterodimeric partner of other NRs including the peroxisome proliferator-activated receptor γ (PPARγ). We previously reported that, in breast cancers (BC), the subcellular localization of these two receptors was strongly associated with patient prognosis. In the present work, we investigated the prognosis value of the combined cytoplasmic expression of RXRα and PPARγ using a retrospective cohort of 250 BC samples. Patients with tumors expressing both NRs in tumor cell cytoplasm exhibited a significant shorter overall (OS) and disease-free survival (DFS). This was also observed for patients with stage 1 tumors. Cox univariate analysis indicated that patients with tumors coexpressing RXRα and PPARγ in the cytoplasm of tumor cells have a decreased 5 y OS rate. Cytoplasmic co-expression of the two NRs significantly correlated with HER2 positivity and with NCAD and CD133, two markers of tumor aggressiveness. Finally, in Cox multivariate analysis, the co-expression of RXRα and PPARγ in the cytoplasm appeared as an independent OS prognosticator. Altogether, this study demonstrates that the cytoplasmic co-expression of RXRα and PPARγ could be of relevance for clinicians by identifying high-risk BC patients, especially amongst those with early and node-negative disease.

## 1. Introduction

Breast cancer (BC) is the most common malignant tumor in women. One of the greatest challenges for clinicians remains to control BC invasiveness and dissemination in order to improve patient survival [1,2]. Indeed, although the treatment options of BC with surgery, chemotherapy, aromatase inhibitors, hormone receptor modulators, and anti-Human Epidermal Growth Factor Receptor 2 (HER2)-Therapy [3,4,5] have developed tremendously in recent years, consistently high mortality rates due to tumor metastasis remain [6,7].

Therapies targeting nuclear receptors (NRs) such as the estrogen receptor (ER) and the progesterone receptor (PR) are highly successful personalized BC therapies used both for prevention and treatment [8]. NRs are activated by binding lipophilic hormones and function mainly as transcription factors in the nucleus [9,10]. In addition to the well-established ER and PR, other NRs, including retinoid X receptor (RXRα), peroxisome proliferator-activated receptor (PPAR), vitamin D receptor (VDR), and others, play notable roles in the pathophysiology of BC and other cancer entities [11,12,13].

In recent studies, we were able to show that cytoplasmic RXRα expression was significantly correlated with a poor outcome in terms of both overall survival (OS) and disease-free survival (DFS) [14]. In addition, another study by our group described that PPARγ was expressed in almost 60% of our breast cancer panel samples with a predominant cytoplasmic location and that high cytoplasmic PPARγ expression was correlated with short OS in breast cancer patients [15].

Because the interplay between these two NRs was not known concerning the prediction of survival for BC patient, the aim of this study was to reanalyze the combined cytoplasmic expression of RXRα and PPARγ in a retrospective cohort of 250 BC samples and describe its association with various clinical parameters including patient survival.

## 2. Materials and Methods

### 2.1. Patients

The data of all included cases (n = 310, 5 cases from patients with bilateral BC) in this retrospective analysis were obtained from the Department of Obstetrics and Gynecology of the Ludwig-Maximilians-University (LMU) Munich, Germany, between 2000 and 2002. No preselection regarding patient or clinical characteristics was made; therefore, it is a representative collective of treated cases in this facility and informed consent to participate was given by all patients. This research was approved by the Ethical Committee of the Medical Faculty, LMU Munich, Germany (approval number 048-08; 18 March 2008). Clinical information was retrieved from the Munich Cancer Registry and samples were encoded during experiments and statistical analysis. All tumors were evaluated according to Union for International Cancer Control (UICC) TNM classification, including pathological tumor size (pT), lymph node involvement (pN), and distant metastasis (M). Grading of the tumor was assessed according to a modification of Elston and Ellis grading proposed by Bloom and Richardson. ER, PR, HER2, and Ki67 were determined at first diagnosis using immunohistochemistry [16].

### 2.2. Immunohistochemical Stainings

Immunohistochemical staining for RXRα and PPARγ was previously described [14,15]. Briefly, sections were first cut and prepared from paraffin-embedded BC samples using standard protocols, followed by incubation in blocking solution (ZytoChem Plus HRP Polymer System Kit, ZYTOMED Systems GmbH, Berlin, Germany). Hereafter incubation with primary antibodies against RXRα [14] (PP-K8508-00, Perseus Proteomics Inc., Tokyo, Japan) or PPARγ [15] (ab59256, Abcam, Cambridge, UK) with a 1:100 dilution for 16 h at 4 °C was performed. Following incubation with a biotinylated secondary anti-rabbit IgG antibody and the associated avidin-biotin-peroxidase-complex (both Vectastain Elite ABC Kit; Vector Laboratories, Burlingame, CA, USA), visualization was performed with 3,3-diamino-benzidine (DAB; Dako, Glostrup, Denmark). Counterstaining of sections with acidic hematoxylin and immediately mounting with Eukitt (Merck, Darmstadt, Germany) was followed by manual analysis with a Diaplan light microscope (Leitz, Wetzlar, Germany) with 25× magnification (see Appendix A for precise protocols). A digital CCD camera system (JVC, Tokyo, Japan) was used for obtaining pictures.

### 2.3. Immunoreactive Score (IRS)

All slides were analyzed in total as large area tissue cuts by two independent blinded observers. The scoring between the two observers differed in 4 cases (n = 1.6%). These cases were re-evaluated by both observers together. After the re-evaluation, both observers came to the same result. The concordance before the re-evaluation was 98.4%. The cytoplasmic staining of RXRα and PPARγ were assessed according to a semiquantitative immunoreactive score (IRS), determined by multiplication of the positive cell proportion score (0 = 0%, 1 = 1–10%, 2 = 11–50%, 3 = 51–80%, and 4 = 81–100% stained cells) and the score of staining intensity (0 = negative, 1 = weak, 2 = moderate, and 3 = strong). For all other markers, staining and IRS were determined without differentiation of nuclear and cytoplasmic staining. A total of 100 cells (3 spots with around thirty cells each) were analyzed for each sample and the IRS corresponded to the mean of the IRS determined on the three spots by the two independent blinded observers.

### 2.4. Statistical and Survival Analysis

To divide the group into cytoplasmic RXRα negative and positive tumors, a cut-off IRS value of 0.5 was used. Receiver operating characteristic (ROC) curve analyses were performed to calculate the optimal cut-off values between low and high PPARγ expression, based upon the maximum differences of sensitivity and specificity as previously described (3.5 was determined as the cut-off for cytoplasmic PPARγ). Correlation analyses were performed by calculating the Spearman’s-Rho correlation coefficient (*p* values of Spearman’s–Rho test presented).

Kaplan–Meier analyses were used to compare survival times. The observation period was up to 10 years. DFS and OS differences were tested for significance by using the chi-square statistics of the log rank test. Statistical analyses above were performed using SPSS 25 (IBM SPSS Statistics, IBM Corp., Armonk, NY, USA). For all analyses, *p* values below 0.05 (*), 0.01 (**), or 0.001 (***) were considered statistically significant. The *p* value and the number of patients analyzed in each group are given for each chart. All risk tables for the different Kaplan-Meier analyses are given in Appendix A.

## 3. Results

### 3.1. Co-Expression of RXRα and PPARγ in BC Tissues

The expression of RXRα and PPARγ was analyzed by immunohistochemistry (IHC) staining in a cohort of 250 BC samples. The detailed clinical characteristics of this cohort are presented in Table 1.

Median age at initial diagnosis was 57.76 years (range from 26.66 to 94.62 years) and the final follow-up time was cut off at 10 years. During this period, 41 cases experienced local recurrences (13.22%) and 57 patients developed distant metastases (18.39%). Finally, 84 cases died (27.09%) and 14 lost follow-up details (4.52%).

As illustrated in Figure 1, both markers were present in the cytoplasm of tumor cells at low or high levels. The expression of both markers in normal breast tissue is shown in Appendix A. As described in the Materials and Method section, for cytoplasmic RXRα, we created two subgroups with negative and positive expression according to the IRS cut-off of 0.5. Using this cut-off, cytoplasmic RXRα staining was observed in 62% (n = 155) of all BC samples. In addition, IRS cut-off of cytoplasmic PPARγ was defined by performing a ROC-curve analysis for 10-year OS (3.5 was determined as the cut-off for cytoplasmic PPARγ). This cut-off divided the whole cohort in PPARγ low expressing group with 119 cases (47.6%) and high expressing group with 131 cases (52.4%).

When we combined the two markers, we were able to define 4 subgroups, as follows: (1) RXRα negative/PPARγ low expression (n = 49); (2) RXRα negative/PPARγ high expression (n = 46); (3) RXRα positive/PPARγ low expression (n = 70); (4) RXRα positive/PPARγ high expression (n = 85). The later (and most relevant subgroup for the present study) with RXRα positive/PPARγ high expression represented 34% of all BC cases.

### 3.2. Cytoplasmic RXRα and PPARγ Expression Correlated with Patient 10-Year OS

Interestingly, as shown in Figure 2A, Kaplan–Meier analyses demonstrated a significant difference (*p* = 0.031) for patient 10-year OS among these 4 subgroups. The curves illustrated that RXRα positive/PPARγ high subgroup specifically correlated with a poor outcome. The comparison between the RXRα positive/PPARγ high subgroup and the other 3 subgroups is visualized in Figure 2B, showing that the 85 cases with both RXRα and PPARγ cytoplasmic expression were significantly associated with shorter 10-year OS (*p* = 0.004). Another way to present the results is shown in Appendix A. When compared to the whole RXRα positive BC subgroup, PPARγ low expression improved the prognosis of these patients with RXRα positive tumors, while PPARγ high expression exhibited an inverse function. In addition, we performed Kaplan–Meier analyses for nuclear staining of RXRα, PPARγ, or combined RXRα/PPARγ, shown in Appendix A. Although there was a trend for a correlation of RXRα nuclear staining with good prognosis, significance was not reached.

Altogether, these data demonstrated that the relative expression of cytoplasmic RXRα and PPARγ impacted patient survival in BC, with positive RXRα/high PPARγ being a factor for poor outcome.

### 3.3. Correlation with DFS in the Whole Cohort and Subgroups

Kaplan–Meier analyses were also performed with the DFS of patients. Focusing on 4 subgroups with different RXRα and PPARγ expression, the positive RXRα/high expression PPARγ subgroup revealed a strong and significant negative impact on 6-year DFS in the whole cohort (Figure 3A, *p* = 0.000046). The similar impaired outcome of positive RXRα/high expression PPARγ was also observed in survival analysis with other 3 subgroups together (Figure 3B, *p* = 0.000002). Moreover, to further analyze this correlation with survival, the whole cohort was stratified according to ER status. The negative influence of positive RXRα/high expression PPARγ on 6-year DFS was also observed for patients with ER-positive tumors (Figure 3C, *p* = 0.000045). Most importantly, we also observed a highly significant correlation with the DFS of patients with stage 1A disease, i.e., with small tumors which had not spread outside the breast (Figure 3D, *p* = 0.000091).

### 3.4. Correlation of Cytoplasmic Positive RXRα/High Expression PPARγ with Clinical Parameters

Tested by Spearman’s–Rho analysis, we analyzed the correlation between positive RXRα/high expressionPPARγ and various relevant clinicopathological characteristics, consist of age, tumor size (pT), lymph node status (pN), tumor grade, ER, PR, Her2, Ki-67, focality status. We also took into account the expression of two aggressiveness markers, CD133, a widely used marker for isolating cancer stem cell (CSC), and N-Cadherin (NCAD), a well-known marker for epithelial-to-mesenchymal transition (EMT) [17]. By analyzing the whole cohort (Table 2) as well as the stage 1 subgroup (Appendix A), positive RXRα/high expression PPARγ was positively correlated with aggressive factors, such as Her2, CD133, and NCAD. No significant associations were observed between positive RXRα/high expression PPARγ and all other parameters.

### 3.5. Cytoplasmic Expression of RXRα/PPARγ as an Independent Prognostic Factor

We first used univariate Cox regression to confirm the trend of poorer survival in patients with positive RXRα/high expression PPARγ (*p* = 0.005, HR: 1.976, 95% CI: 1.232–3.168) with 5- or 10-year OS rate of 69.41% and 31.76%, respectively (Table 3). In contrast, negative RXRα/low PPARγ in the whole cohort (*p* = 0.086, HR: 0.541, 95% CI: 0.268–1.090) was related to a favorable prognosis with 5- or 10-year OS rate of 79.59% and 51.02%, respectively.

Multivariate analysis was also performed for the whole cohort, using a Cox regression model with cytoplasmic positive RXRα/high expression PPARγ and 6 relevant clinicopathological parameters, namely histology, pT, pN, Grading, ER, and PR (Table 4). Among clinical features, only pN and ER were independent prognostic markers of 10-year OS. Of note, combinational RXRα positive/PPARγ high expression was also regarded as an independent prognostic marker, with a hazard ratio of 2.331 (95% CI: 1.147–4.736) and a strong significance (*p* = 0.019).

Interestingly, as shown in Table 5, when multivariate analysis was performed in the stage 1A subcohort (using a Cox regression model with the same 7 parameters except pN), cytoplasmic coexpression of RXRα and PPARγ was again an independent prognostic marker of 10-year OS, with a very high significance (*p* = 0.000079).

## 4. Discussion

Within this study, we analyzed the prognosis value of the combined cytoplasmic expression of RXRα and PPARγ in a retrospective cohort of BC patients. Patients with tumors expressing both NRs in tumor cell cytoplasm exhibited a significant shorter overall and disease-free survival and this was also observed for patients with small and localized tumors. Cox univariate analysis indicated that patients with tumors expressing cytoplasmic RXRα and PPARγ had a decreased 5 year-OS rate. Cytoplasmic co-expression of the two NRs significantly correlated with HER2 positivity and with NCAD and CD133, two markers of tumor aggressiveness. Finally, in Cox multivariate analysis, the co-expression of RXRα and PPARγ in the cytoplasm appeared as an independent negative OS prognosticator.

Retinoid X receptors and especially RXRα preferentially form heterodimers with PPARγ bound in the nucleus at specific response elements (proliferator-activated receptor response elements or PPREs) present in target gene promoters [18]. Following activation by its ligands 15d-PGJ2, PPARγ heterodimerizes with RXRα and exerts antigrowth effects in cancer cells [19,20,21,22]. The RXRα/PPARγ heterodimer has also been shown to bind directly to nuclear estrogen response elements (ERE) independently of estrogen receptor (ER) activity [23]. The authors propose to evaluate, in postmenopausal women after primary surgery, the effect of combining an RXR-selective retinoid with either troglitazone or a long-chain n-3 polyunsaturated fatty acid (PUFA) on possible tumor progression. The present study indicates that cytoplasmic localization of PPARγ is needed to observe a significant correlation of cytoplasmic RXRα with poor prognosis (see Figure 2A). One can speculate that the RXRα/PPARγ heterodimer exerts specific functions in the cytoplasm, as it has been suggested for PPARs [24].

In a recent study published by our group, investigations on RXRα as a single survival factor revealed a trend association of its nuclear expression with improved OS [14]. Cytoplasmic RXRα, on the other hand, was significantly correlated with a worse prognosis [14]. Interestingly, the different BC sub cohorts in our patient collective showed a significant decrease in OS for triple-negative and HER-2 neu-negative BC tissue that stained positive for cytoplasmic RXRα. The hypothesis that the shuttling of RXRα out of the nucleus, or RXRα knock-out, may lead to a deterioration of survival is supported by these results [14]. We also reported a correlation between high cytoplasmic PPARγ expression and short OS in general. However, the bad prognostic impact of cytoplasmic PPARγ depends on Cox-1 expression because cytoplasmic PPARγ expression was an independent prognostic parameter of OS only in N-cadherin low and Cox-1 negative tumors [15].

In another study, we were able to show that RXRα and PPARγ are overexpressed in BRCA1 mutated breast cancer cases and predict prognosis [25]. In BRCA1 mutated cases, RXRα was expressed with a prominent nuclear pattern, while PPARγ staining was mainly located in the cytoplasm [25]. Via sumoylation and ubiquitinilation, the BRCA1 protein participates in the degradation of RXR and PPARγ and other nuclear receptors [26,27,28] and this may explain the overexpression of these receptors in BRCA1 mutated genetic backgrounds [25].

It should be stated that combined analyses of both receptors on tumor development and prognosis are rare. In an early study, Desreumaux et al. showed a protective effect against colon inflammation through activation of the RXRα/PPARγ heterodimer and suggested that, based on their synergistic actions, the combination of RXRα and PPARγ ligands could be a promising therapeutic approach [29]. Another study showed that human multiple myeloma cells expressed PPARγ and underwent apoptosis upon exposure to PPARγ ligands. Interestingly, the RXRα ligand 9-cis retinoic acid (9-cis RA) in combination with PPARγ ligands greatly enhanced multiple myeloma cell killing [30]. Giaginis et al. investigated the association between RXRα and PPARγ expression with clinicopathological parameters, tumor proliferative capacity and patients’ survival in pancreatic ductal adenocarcinoma [31]. PPARγ staining intensity was associated with shorter overall survival in univariate analysis and proved to be an independent prognostic factor in multivariate analysis, whereas RXRα failed to predict patient survival [31]. Another study on osteosarcomas showed that activation of RXRα and PPARγ may synergistically inhibit tumor growth and cell proliferation, at least partially by inducing osteoblastic differentiation of osteosarcoma cells [32]. More recently, Chen et al. showed that a combination of RXRα and PPARγ agonists induced sodium/iodide symporter expression and inhibited cell growth of human thyroid cancer cells [33].

In human breast cancer cells, it has been reported that the association of the PPARγ ligand Rosiglitazone and 9-cis retinoic acid as a RXR ligand induces apoptosis [34]. A more recent study showed that RXRα and PPARγ ligands disrupt the inflammatory cross-talk in the hypoxic breast CSC niche [35]. Our own findings on RXRα/PPARγ cytoplasmic coexpression showed that it was significantly correlated with CD133, a widely used marker for cancer stem cell (CSC). Therefore, one can imagine that the relocalization of the two NRs from the cytoplasm into the nucleus upon ligand activation might interfere with the CSC stem cell properties of the cancer cells by reducing their differentiation grade and explain why the nuclear expression of this marker combination can be considered as a positive factor for survival. Future work will be needed to specifically investigate the role of RXRα and PPARγ in the breast CSC biology.

One drawback of the present study is that it is based on a single dataset which may not encompass the full diversity of breast cancer entities. Another limitation is that patient treatments were not taken into account in the clinical parameters as they changed noticeably within the time of this retrospective study. Consequently, further studies using other cohorts will be required to validate and extend our results.

## 5. Conclusions

This study demonstrates that the cytoplasmic co-expression of RXRα and PPARγ is an independent negative prognosticator for BC patients. These results could be of relevance for clinicians by identifying high-risk patients, especially amongst those with early and node-negative disease.

## Figures and Tables

**Figure 1 cells-11-01244-f001:**
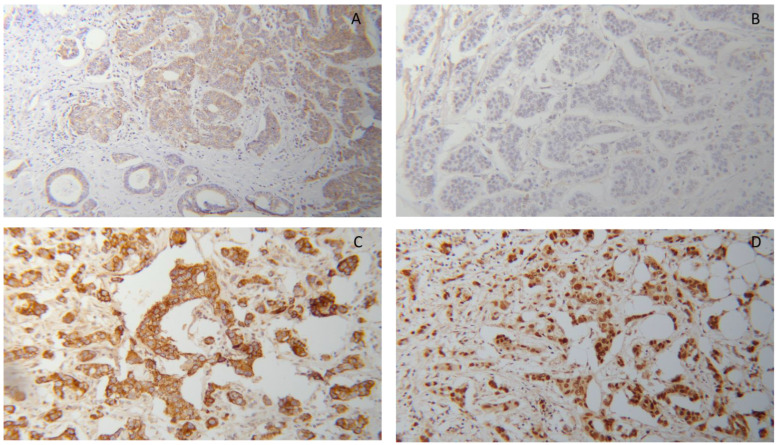
Immunohistochemical detection of PPARγ (**A**,**C**) and RXRα (**B**,**D**) in breast cancer tissues from two patients showing weak (**A**,**B**) or strong (**C**,**D**) expression of PPARγ and RXRα in the same tumor, respectively; all pictures 10× lens.

**Figure 2 cells-11-01244-f002:**
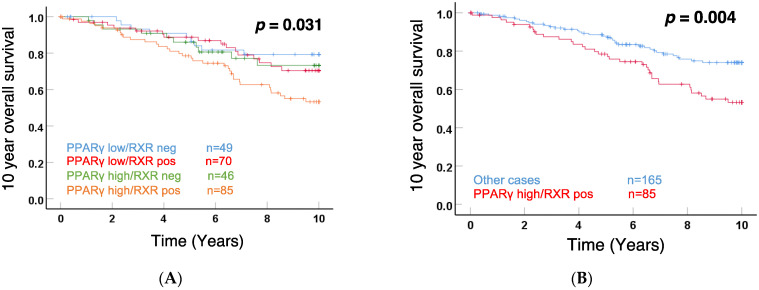
The ten-year OS rates for the 250 breast cancer cases with combined RXRα/PPARγ expression divided into 4 subgroups (**A**) and summarized RXRα positive/PPARγ high expression compared to all other cases (**B**) with indication of the respective *p*-value.

**Figure 3 cells-11-01244-f003:**
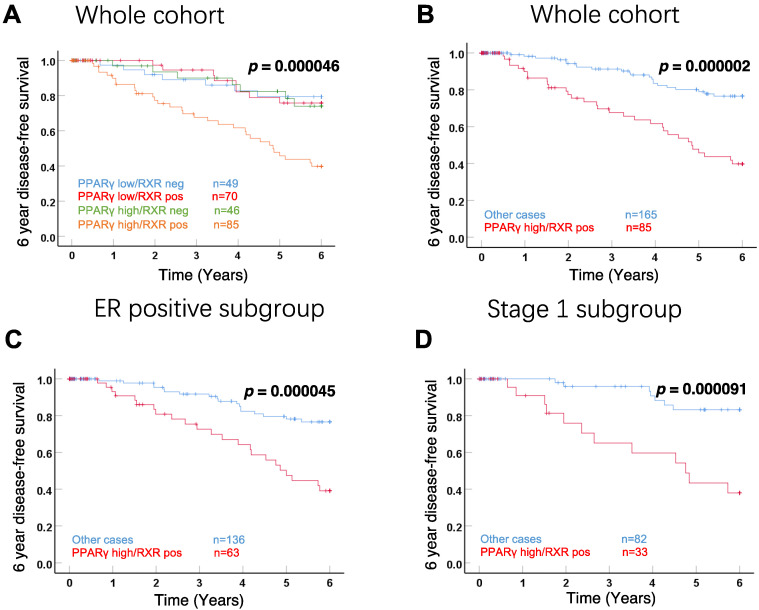
Kaplan–Meier analyses of 6-year disease-free survival for the 4 subgroups with different RXRα and PPARγ expression (**A**), positive RXRα/high expression PPARγ compared to all other cases (**B**) and ER-positive (**C**) or stage 1A (**D**) BCs.

**Table 1 cells-11-01244-t001:** Clinical and pathological characteristics of all patients.

Clinical and Pathological Characteristics ^a^	n = 250 ^b^
Age, median (years)	57.76
Molecular subtype (IHC)	
Luminal A (Ki67 ≤ 14%)	135
Luminal B (Ki67 > 14%)	56
HER2 positive	16
HER2 non luminal	9
Triple negative	33
Unknown	1
Grade	
I	12
II	90
III	41
Unknown	107
Tumor size	
pT1	165
pT2	71
pT3	4
pT4	10
Lymph node metastasis	
Yes	105
No	140
Unknown	5
ER status	
Positive	199
Negative	51
PR status	
Positive	145
Negative	105
Cytoplasmic PPARγ expression	
Low (IRS < 3.5)	119
High (IRS > 3.5)	131
Cytoplasmic RXRα expression	
Negative	95
Positive	155

^a^ All information given refer to the primary tumor; ^b^ 3 of 247 patients are bilateral primary BC, so we deal with the tumor as individual one (n = 250).

**Table 2 cells-11-01244-t002:** Correlations of double cytoplasmic staining (RXRα positive/PPARγ high) with clinical parameters in the whole cohort of 250 BC patients.

	Correlation Coefficient	*p*	n
Age	0.002	0.755	250
pT	0.106	0.094	250
pN	0.05	0.433	245
Grade	−0.029	0.733	143
ER	−0.0099	0.109	261
PR	−0.034	0.580	261
Her2	0.160 **	0.010	261
Triple-negative	0.074	0.231	262
Ki-67	0.020	0.773	203
Focality	0.035	0.572	262
CD133	0.172 **	0.007	244
NCAD	0.262 ***	0.000036	243

Significant *p*-values are shown by ** (*p*-value < 0.01) and *** (*p*-value < 0.001).

**Table 3 cells-11-01244-t003:** Cox univariate survival analyses for hazard ratio (HR) determination and *p* values in the whole cohort of BC patients.

Group	*p*	HR	95% CI	5-Year OS Rate	10-Year OS Rate
RXRα neg/PPARγ low	0.086	0.54	0.26–1.09	79.59%	51.02%
RXRα neg/PPARγ high	0.507	0.79	0.40–1.37	71.74%	32.61%
RXRα pos/PPARγ low	0.394	0.78	0.44–1.37	71.43%	35.71%
RXRα pos/PPARγ high	0.005 **	1.97	1.23–3.16	69.41%	31.76%

Significant *p*-values are shown by ** (*p*-value < 0.01).

**Table 4 cells-11-01244-t004:** Cox multivariate analysis for positive RXRα/high expression PPARγ in the whole cohort.

	*p*	HR	95% CI
Histology	0.225	1.018	0.989–1.047
pT	0.066	1.252	0.985–1.590
pN	0.016 *	1.269	1.046–1.541
Grading	0.414	1.363	0.648–2.865
ER	0.242	0.495	0.153–1.607
PR	0.810	1.117	0.453–2.750
Her2	0.167	2.197	0.720–6.708
Triple-negative	0.627	1.489	0.299–7.416
PPARγ high/RXR pos	0.035 *	2.175	1.054–4.487

Significant *p*-values are shown by * (*p*-value < 0.05).

**Table 5 cells-11-01244-t005:** Cox multivariate analysis for positive RXRα/high expression PPARγ high expression in the stage 1A cohort.

	*p*	HR	95% CI
Histology	0.233	0.833	0.618–1.124
pT	0.269	4.441	0.315–62.589
pN	-	-	-
Grading	0.332	0.454	0.092–2.238
ER	0.521	0.364	0.017–8.004
PR	0.657	1.810	0.132–24.726
Her2	0.185	7.835	0.374–163.958
Triple-negative	0.487	4.336	0.069–272.193
PPARγ high/RXR pos	0.013 *	10.065	1.644–61.617

Significant *p*-values are shown by * (*p*-value < 0.05).

## Data Availability

Data are available upon request from the corresponding author due to ethical reasons.

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
