# Peer review of "Cytoplasmic Colocalization of RXRα and PPARγ as an Independent Negative Prognosticator for Breast Cancer Patients"

_cells, 2022, doi:10.3390/cells11071244_

Round 1

Reviewer 1 Report

Review of the manuscript untitled « Cytoplasmic colocalization of RXRα and PPARγ as an

independent negative prognosticator for breast cancer patients »

In a retroscpective cohort of breast cancer specimen the authors studied the expression of 2 nuclear receptors : RXRα and PPARγ by IHC. The pape ris well writtren and the results are interesting and clearly presented.

I have two concerns before publication :

  • I would like to see pictures showing high and low staining for both receptors.
  • The staining shows also a nuclear staining not mentioned. What about the correlation with clinical outcome ?

Author Response

Reply to reviewer#1

In a retrospective cohort of breast cancer specimen the authors studied the expression of 2 nuclear receptors : RXRα and PPARγ by IHC. The paper is well writtren and the results are interesting and clearly presented.

 I have two concerns before publication:

  1. I would like to see pictures showing high and low staining for both receptors.

Answer: We included pictures showing weak and strong staining of both receptors in the revised Figure 1A-D.

  1. The staining shows also a nuclear staining not mentioned. What about the correlation with clinical outcome?

Answer: We performed Kaplan-Meier analyses for nuclear staining of PPARg, RXRa or combined PPARg/RXRa. Although there is trend for a correlation of RXRa nuclear staining with good prognosis, significance was not reached. All three curves have been included as Supplementary Figure 2A-C

Reviewer 2 Report

The manuscript, ‘Cytoplasmic colocalization of RXRα and PPARγ as an independent negative prognosticator for breast cancer patients, ‘is overall, well written and presented in a well-structured manner.  The study demonstrates that the cytoplasmic co-expression of RXRα and PPARγ may be beneficial for identifying high-risk BC patients.

My comments are as follows:

  1. What the expression level of RXRα and PPARγ in normal breast tissue was observed?
  2. All the tissues used for immunohistochemistry are full-face. Please clarify?
  3. How the authors validated the specificity of the RXRα and PPARγ is not clear, so would be better to add how they evaluated the specificity of the antibody. Antigen retravel, controls used, should be included in the main results or as supplementary data, along with dilutions and specifications of the secondary antibody.
  4. For scoring is more than one observer evaluating the intensity and distribution pattern of the staining? If so

What was the level of discordance between the two observers for scoring?

  1. Inter-observer agreement and the interclass correlation coefficient need to be included to evaluate the concordance between the 2 scorers

  1. KM figures - The authors should consider adding risk tables below the KM curves to give a better sense of the group’s sizes and follow-up.

  1. Their multivariate analysis using the Cox-regression model (Table 4 and 5) does not include major confounding in Breast Cancer, subtypings. If they really want to show the relevance of expression, should’ve included subtyping of BC multivariate analysis.
  2. Limitation of the study better to be included in the discussion

Author Response

Reply to reviewer#2

The manuscript, ‘Cytoplasmic colocalization of RXRα and PPARγ as an independent negative prognosticator for breast cancer patients, ‘is overall, well written and presented in a well-structured manner.  The study demonstrates that the cytoplasmic co-expression of RXRα and PPARγ may be beneficial for identifying high-risk BC patients.

My comments are as follows:

  1. What the expression level of RXRα and PPARγ in normal breast tissue was observed?

Answer: We have included pictures showing the staining of both nuclear receptors in normal glandular breast tissue as Supplementary Figure 1A-B. PPARg expression seems to be rather weak and RXRa expression appears heterogeneous, which is in concordance with data from the Human Protein Atlas (see HPA PPARg and HPA RXRa).

  1. All the tissues used for immunohistochemistry are full-face. Please clarify?

Answer: We analyzed all slides in “full-face” (large area tissue cut). This has been added in the Material and Method section on lines 94-95.

  1. How the authors validated the specificity of the RXRα and PPARγ is not clear, so would be better to add how they evaluated the specificity of the antibody. Antigen retravel, controls used, should be included in the main results or as supplementary data, along with dilutions and specifications of the secondary antibody.

Answer: We have included the staining protocols with all data and control staining pictures as Supplementary Table 1.

  1. For scoring is more than one observer evaluating the intensity and distribution pattern of the staining? If so what was the level of discordance between the two observers for scoring? Inter-observer agreement and the interclass correlation coefficient need to be included to evaluate the concordance between the 2 scorers

 Answer: The intensity and distribution pattern of the immunochemical staining reaction were evaluated by two independent blinded observers. In 4 cases (n = 1.6%), the evaluation of the two observers differed. These cases were re-evaluated by both observers together. After the re-evaluation, both observers came to the same result. The concordance before the re-evaluation was 98.4%. This precision was added in the paragraph 2.3 in the Materials and Methods section (lines 97-100).

  1. KM figures - The authors should consider adding risk tables below the KM curves to give a better sense of the group’s sizes and follow-up.

Answer: We have included these data in the Supplementary Tables 2 - 6.

  1. Their multivariate analysis using the Cox-regression model (Table 4 and 5) does not include major confounding in Breast Cancer, subtypings. If they really want to show the relevance of expression, should’ve included subtyping of BC multivariate analysis.

Answer: We have included the HER2/triple negative subtype (which has the worse prognosis) as one of the parameters in the multivariate Cox-regression analysis for either the whole cohort or the stage 1 subgroup. In both cases, the p value remains significant for PPARγ high/RXR pos (p=0.035 in the whole cohort and p=0.013 in the Stage 1 subgroup of patients). Table 4 and 5 now contain the HER2/triple negative subtype.

  1. Limitation of the study better to be included in the discussion

Answer: We have included few sentences on the limitations of the study in the Discussion section of the revised manuscript.
